# Blockchain Based Smart-Grid Stackelberg Model for Electricity Trading and Price Forecasting Using Reinforcement Learning

Md Mahraj Murshalin Al Moti [1,†], Rafsan Shartaj Uddin [1,†], Md. Abdul Hai [1,†], Tanzim Bin Saleh [1,†], Md. Golam Rabiul Alam [1,†], Mohammad Mehedi Hassan [2,*,†] and Md. Rafiul Hassan [3,†]

1   Department of Computer Science and Engineering Brac University, Dhaka 1212, Bangladesh; md.mahraj.murshalin.al.moti@g.bracu.ac.bd (M.M.M.A.M.); rafsan.shartaj.uddin@g.bracu.ac.bd (R.S.U.); md.abdul.hai@g.bracu.ac.bd (M.A.H.); tanzim.bin.saleh@g.bracu.ac.bd (T.B.S.); rabiul.alam@bracu.ac.bd (M.G.R.A.)
2   Information Systems Department, College of Computer and Information Sciences, King Saud University, Riyadh 11543, Saudi Arabia
3   College of Arts and Sciences, University of Maine, Presque Isle, ME 04769, USA; md.hassan@maine.edu
*   Correspondence: mmhassan@ksu.edu.sa
†   These authors contributed equally to this work.

**Abstract:** A smart grid is an intelligent electricity network that allows efficient electricity distribution from the source to consumers through telecommunication technology. The legacy smart grid follows the centralized oligopoly marketplace for electricity trading. This research proposes a blockchain-based electricity marketplace for the smart grid environment to introduce a decentralized ledger in the electricity market for enabling trust and traceability among the stakeholders. The electricity prices in the smart grid are dynamic in nature. Therefore, price forecasting in smart grids has paramount importance for the service providers to ensure service level agreement and also to maximize profit. This research introduced a Stackelberg model-based dynamic retail price forecasting of electricity in a smart grid. The Stackelberg model considered two-stage pricing between electricity producers to retailers and retailers to customers. To enable adaptive and dynamic price forecasting, reinforcement learning is used. Reinforcement learning provides an optimal price forecasting strategy through the online learning process. The use of blockchain will connect the service providers and consumers in a more secure transaction environment. It will help tackle the centralized system's vulnerability by performing transactions through customers' smart contracts. Thus, the integration of blockchain will not only make the smart grid system more secure, but also price forecasting with reinforcement learning will make it more optimized and scalable.

**Keywords:** smart grid; blockchain; price forecasting; electricity demand and supply; smart meter; reinforcement learning; Stackelberg model

## 1. Introduction

A smart grid is superior to a traditional grid because it understands the consumption and distribution of energy. Traditional grids tend to waste a lot of energy while generating and distributing electricity. Smart grids add telecommunication features to a traditional grid to make energy demand and supply more efficient and reliable [1–4]. Consumers at home have smart meters that monitor the usage of power and send the data to power generators to respond to their demands. It monitors when the usage of power is high and low, so that power plants can better direct electricity when only necessary and reduce wastage. The traditional implementation of a centralized smart grid system is vulnerable as a single main point of failure can shut down a city. A centralized system is highly prone to malicious attacks as customer usage, and billing information may be compromised. This can be solved with the help of blockchain, where information is stored on the computers of consumers [5].

Demand response (DR) is an efficient method to use in a smart grid system to reduce costs and improve grid efficiency. Price-based DR refers to influencing the customers' electricity usage with the variable electricity prices. To help with dynamic pricing in a hierarchical energy market, this paper proposes a DR algorithm that keeps up with the dynamic pricing while also reducing the service provider's (SP) and customer's (CU) costs. With the help of reinforcement learning (RL) and *Q*-Learning, we are predicting the price of electricity. The SPs possess the flexibility to set the price dynamically in accordance with demand and level of dissatisfaction. Furthermore, blockchain integration of every customer's profile will help to secure the decentralized transaction of electricity between SP and CU.

In DR efficiency, we can see several works regarding DR models that help to subsidize customer usage by minimizing costs. There is one such example in [6–8], where electricity consumption of different home appliances was monitored, and time-of-use (TOU) pricing helped to minimize customers' costs. In [9–14], we can see the benefit of predetermined next-day electricity prices and efficient scheduling to help keep the costs of CU in check. Though the contributions of the papers mentioned above helped the electricity demand response field to a good extent, it still lacks the dynamic market where demand is ever-changing. Thus, a DR strategy with dynamic pricing compatibility is apt for modern usage. Another smart way to provide a variable amount of service to customers is by using a dynamic pricing model, which changes the price of energy from time to time for perfect resource allocation [15]. One of the implementations can be found where retailer profit has been maximized with a quadratic programming problem [16]. The deterministic dynamic pricing approach and abstract linear models are sometimes unable to provide optimal performance due to any variable change in demand, resulting in loss of money, and secondly, abstract models visualize an approximation of energy distribution and are more dependent on the modeler's experience. This is where reinforcement is a viable solution, as it is model-free and can efficiently react to the ever-changing demand for CUs and benefit both CUs and SPs. Therefore, the key contributions of this research are as follows:

- This research introduced a dynamic Stackelberg model-based retail price forecasting of electricity in a smart grid. The Stackelberg model considered two-stage pricing between electricity producers to retailers and retailers to customers. To enable adaptive and dynamic price forecasting, reinforcement learning is used.
- A blockchain-based electricity marketplace is proposed for the smart grid environment to enable a decentralized ledger in the electricity market.
- The blockchain-based smart grid electricity marketplace is implemented, and the simulation of the system returns responsive retail prices, a change in energy consumption due to change in price, and the price pattern for an entire day. Moreover, it assesses the quality and performance of the dynamic pricing system for the demand response program. The simulation of an entire day for each customer shows that the retail price never falls below the wholesale price; however, it also changes to a price as close to the wholesale price when dissatisfaction of customers is at the maximum due to a rise in demand at lower consumption rates. Therefore, the simulation shows that the prices are responsive for both retailers and customers.

The rest of the paper is organized as follows: Section 2 contains a literature review and an overview of the related works. Section 3 contains the system architecture and proposed blockchain model. Section 4 describes the *Q*-learning method for smart grid price forecasting, which includes the customer model, service provider model, *Q*-learning-based price forecasting and our proposed *Q*-learning algorithm. The implementation of blockchain and implementation of *Q*-learning-based price forecasting are shown in Section 5. Finally, Section 6 is about the conclusion, which contains a research overview, contribution, impacts, and future works.

## 2. Literature Review

A smart grid is a new system of power distribution, and there is already numerous research on the subject. However, with new technology comes quite a few shortcomings. Price forecasting and price forecasting accuracy are closely related to how the demand and supply will turn out. Wang et al. suggested a multi-layered neural network for price forecasting [17]. However, with such a complex network, we gain a high computational time; moreover, the work proposed by M. Zahid et al. shows the loss of neurons is also very high [5].

Lago et al. suggested several price forecasting methods using deep learning techniques. The study proposes deep neural networks with a hybrid long short-term memory and deep neural network structure to significantly improve prediction accuracy. However, that paper is only compared using only a single dataset. Therefore, it is not suitable to use such a paper for real-life experiments and appliances since there are so many factors to consider [18].

Yu et al. [19] presented a price-based novel demand-response (DR) model to enhance the electricity management system between the utility company and the consumers. Using a pricing function, the real-time price (RTP) is manipulated, and the balance of supply and demand is obtained. Here, the Stackelberg game model is used to flatten the system's aggregated loads while maintaining the utility company's benefit and the user's cost minimization. Though the proposed model achieved the lowest peak-to-average ratio (PAR) and highest load factor (LF), the system could not provide an optical communication network, and transactions were not secured or recorded. Moreover, lower computational time can be obtained using different models.

Jiang et al. [20] proposed a fair transaction packing strategy for permissioned blockchains using IoT systems based on transaction response time. The FAIR-PACK algorithm uses a heuristic and a min-heap technique to divide the data into two subset sum problems for different variables. Extensive comparisons of time and performance complexity are performed to see how the transaction arrival rate, block generation time, block size, and block validity ratio impact FAIR-PACK performance. The analysis revealed that FAIR-PACK delivers more fairness and a faster average reaction time than prior studies since it uses a novel technique to conduct the transactions.

The searchable blockchain system can allow an accurate search over encrypted distributed storage systems and protect privacy. Although, it comes with limitations as only single-keyword searches are logically functional within the system, and applying multi-keyword searches concerns confidentiality and efficiency of the database. Jiang et al. [21] suggested an advanced blockchain-based framework for information systems that can perform dynamic modifications and multi-keyword searches, ensuring preserved privacy. The bloom filter determines a low-frequency keyword from the database to operate search instructions providing reduced computational costs and search space. It also computes the operation in a single iteration, resulting in complete privacy protection. According to the outcomes, the multi-keyword search protocol exceeds the existing technique by 14.67 percent for the time delay and 59.96 percent for financial expenses. Tuning the parameters of the filter system can bring about further improvements and provide a more robust system.

Blockchain technology plays a vital role in securing the vast healthcare system. Jiang et al. [22] suggested a framework named BlocHIE, a blockchain tool for sharing healthcare data. They analyze the sharing structure and store the data into two distinct blockchains; EMR-Chain for electronic medical records and PHD-Chain for personal healthcare data. It abolishes the issues of the traditional storing system by employing on-chain verification, ensuring immutability and anonymity. Moreover, they apply FAIR-FIRST and TP&FAIR, two fairness-based packing techniques that increase system capacity and user fairness to ensure privacy-preserving.

Lahouar et al. [23] discussed electrical forecasting, which is becoming increasingly popular due to the deregulation and integration of renewable resources. Because of the limited power source, it is crucial to predict the power prediction accurately. Therefore, this paper proposes a short-term load predictor that can forecast for the next 24 h of loading

for predicting power consumption. They used ANN (artificial neural networks) and SVM (support vector machines) technology in their project. Tunisian Power Company tests its proposed system to produce accurate and acceptable results one day in advance, with an average error rarely exceeding 2.3%. However, their computational time is very high.

## 3. Proposed Blockchain-Based Model for Electricity Trading in Smart Grid

### 3.1. System Architecture

A hierarchical electricity market of a state or city is taken into consideration. The three key components of this market are grid operator, service provider and customer. Grid operators operate the electricity in the smart grids and are also responsible for the installation and management of smart grids. High voltage electricity in the smart grids is then transferred to the service providers by the grid operator. GO sells the nationwide electricity at a wholesale market price. This electricity is then transferred to CUs by the SPs at a low-voltage stage. This dynamic price is determined by the SPs [24]. This determining pricing strategy by the SPs is what is mainly focused on in this paper for price forecasting. With efficient energy consumption and profit maximization in mind, SPs need to decide on the dynamic retail price for the highest benefit for both sides. CUs cooperate with them to adjust their usage and reduce costs from time to time. The SPs have the load profile information of customers as well as information about the purchase from GOs to help them adaptively change the prices.

Figure 1 shows an overview of the entire system and how the parties are interconnected with each other. Several energy sources are considered producers that generate a vast amount of energy. It is distributed among the consumers by the service provided. All this information is stored in a distributed network, and the transactions are recorded in it as well.

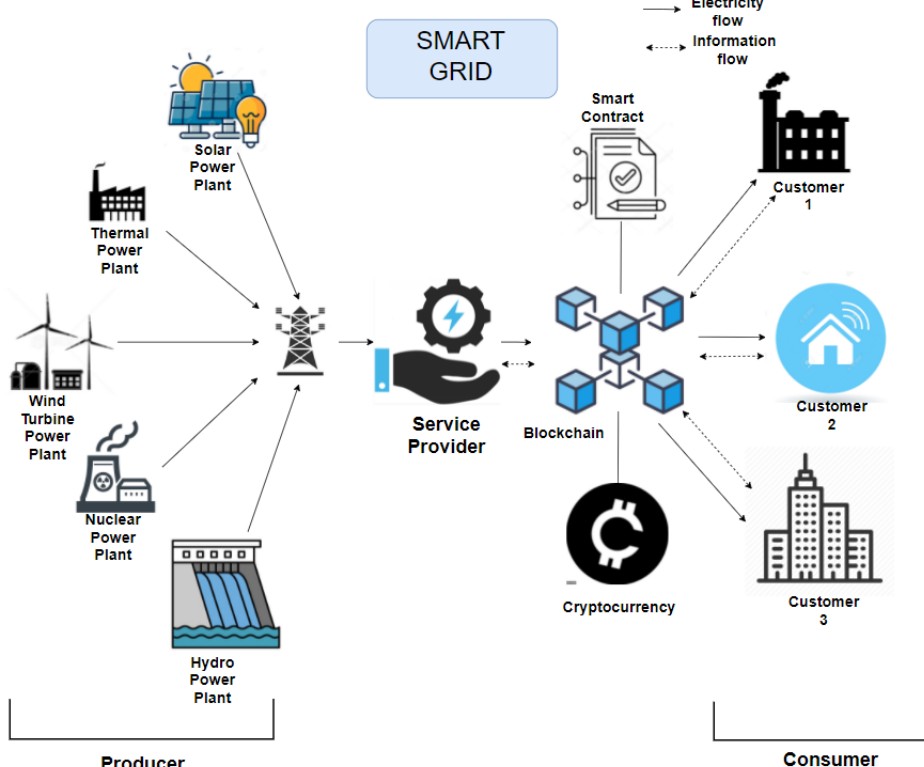

**Figure 1.** System model of blockchain-based Stackelberg smart grid environment.

### 3.2. Smart Contract

A smart contract is a set of rules and agreements approved by both parties. It offers authenticity to verify the effectiveness of any operation or action. After deploying the smart contract, some functions and events are performed to confirm the transaction. It can assist us in transferring money, transferring assets and shares, and other significant transactions in a transparent manner.

How Smart Contract Works

A smart contract acts as a third party in the blockchain and involves dealing and ensuring trust between the buyer and seller. It is an agreement between the buyer and seller with multiple stored conditions in the system and can not be manipulated. It contains three main mechanisms: the parties' contractual agreements, the administration of predetermined conditions essential for the contractual responsibilities to be fulfilled, and the deployment of the smart contract [25]. It is a fully decentralized system that does not require any additional party to conduct it; it is immutable, secure, and distributed among all the nodes.

### 3.3. User Layer

The user layer comprises all the users who purchase electricity from the market for their routine work. No central party is needed for communication between providers and consumers; instead, the communication is conducted directly through the system. Blockchain can be used to share information between the provider and consumers. Users provide information to the information layer, connected with the user layer, to become registered in the blockchain network. Figure 2 illustrates the suggested blockchain architecture.

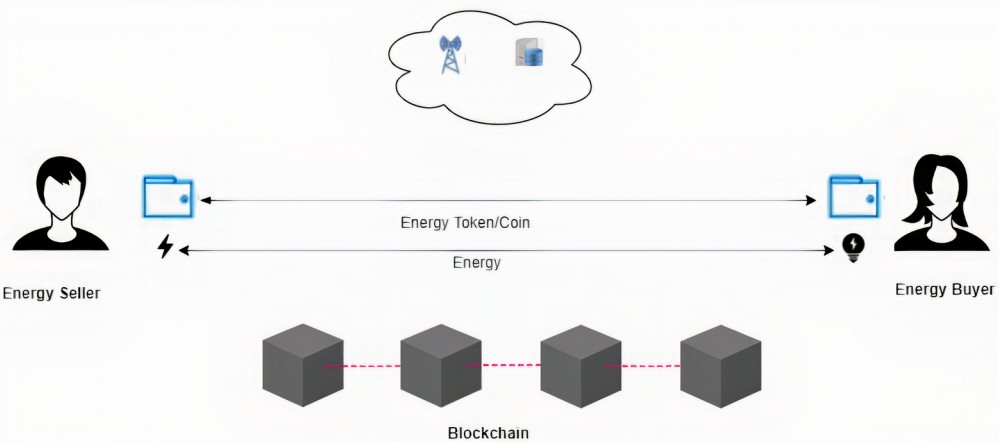

**Figure 2.** Proposed blockchain architecture.

### 3.4. Information Layer

The information layer consists of records and ownership of energy. A buyer or seller must register himself in the smart grid system by providing important information such as name and address. Upon joining the system, a user receives a private ID and profile that shows the user's information, which is stored in the form of a hash in a block of the blockchain. After authenticating, users can view their electricity usage history. It helps to monitor user activity reliably, and the SG stores all information of users in a decentralized system and stores data in encrypted form. Based on the buying information, one can understand how much electricity is needed and if any energy is being wasted without proper usage.

### 3.5. Users Authorization

A smart contract allows users to become a part of the network and check the previous electricity trading history. The users need to register for authorization and authentication, which increases the security issue. For registration into the smart contract, the initial step is

being verified as a seller or a buyer. After the authentication, users can become a part of the electricity trading circuit and can pick trading time and price. When all transactions are completed, a block is created. In any transactions, hashes are calculated to verify the original owner's address. However, even after trading, no one can check each other's name, address, or anything, and only the account's address is visible. Without proper authorization of the transactions, the money will not be transferred, and the trading request will not be accepted; thus, the transaction will not go through.

Figure 3 demonstrates how the functions sequentially work and how the system processes within the blockchain system.

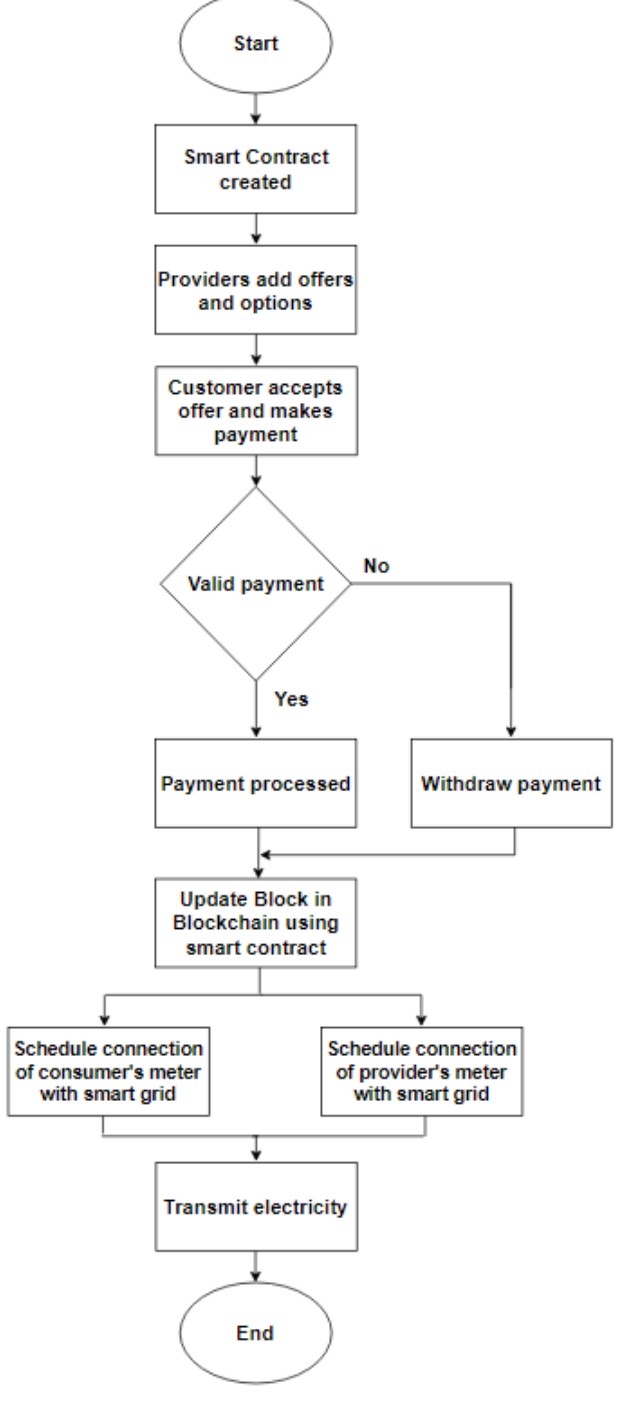

**Figure 3.** The flow diagram of blockchain-based smart grid system.

## 4. Proposed *Q*-Learning Method for Smart Grid Price Forecasting

*4.1. Problem Formulation*

4.1.1. Customer Model

There are two types of energy load profiles that a customer can have. One is the critical load, and the other is the curtailable load. These are classified based on their energy requirement and priorities.

Critical load is the load that is of high priority and must be met at any cost for customer satisfaction and priority. An example can be use of power in data centers and power stations. The equation that satisfies the critical load demand is denoted as follows:

$$ec_{t,c}^{\text{critic}} = ed_{t,c}^{\text{critic}} \tag{1}$$

where $t$ is divided into 24 h segments, which represent each hour of the day. The price will be updated every 24 h. $c \in \{1,2,3\ldots C\}$ represents the customers $c$. $ed_{t,c}$ refers to the energy demand, and $ec_{t,c}$ refers to the energy consumption of a customer $c$ at a specific time $t$.

Curtailable load, on the other hand, is more flexible with price. The demand of customers for curtailable load can change with price as the demand falls with the increase in electricity price. For a certain customer CU $c$, consuming a certain amount of $ec_{t,c}$ at time $t$ will correspond to the customers' amount of load satisfaction. Subtracting that load satisfaction from the total amount of energy demand gives us the dissatisfaction level at time $t$, which is $ed_{t,c} - ec_{t,c}$. This dissatisfaction level is denoted as $\phi_{t,c}$. This signifies the degree of dissatisfaction that customers can experience when prompted to reduce their electricity demand due to high prices. This co-efficient is convex in nature and tends to increase massively if energy reduces significantly. The equation that satisfies the critical load demand is denoted as follows:

$$
\begin{aligned}
&ec_{t,c}^{\text{curt}} = ed_{t,c}^{\text{curt}} \cdot \left(1 + \xi_t \cdot \tfrac{\lambda_{t,c} - \pi_t}{\pi_t}\right) \\
&\xi_t < 0 \\
&\lambda_{l,c} \geqslant \pi_t
\end{aligned}
\tag{2}
$$

Here, $\xi_t, \lambda_{t,c}, \pi_t$ denote elasticity coefficient, retail price for customer $c$ and wholesale price at time $t$, respectively.

In economic terms, elasticity $\xi_t$ measures responsiveness of the change between two variables with respect to one another. Price elasticity of demand can show how the change in product price can impact the energy demand of a particular good. In the context of smart grids, this elasticity refers to the change in the demand for electricity with the 1% increase in price of that particular time. Thus, this elasticity between the demand and price is inversely proportional. Research regarding elasticity on smart grid energy concludes that the demand for electricity is most elastic during peak hours, and long-run elasticity results are better compared to the short one. The elasticity values used in this paper to study the RL implementation were acquired from published papers.

The dissatisfaction cost function can be defined as follows:

$$
\begin{aligned}
&\varphi_{t,c} = \tfrac{\alpha_c}{2}\left(ed_{t,c}^{\text{curt}} - ec_{t,c}^{\text{curt}}\right)^2 + \beta_c\left(ed_{t,c}^{\text{curt}} - ec_{t,c}^{\text{curt}}\right) \\
&\alpha_c > 0 \\
&\beta_c > 0 \\
&D_{\min} < ed_{t,c}^{\text{curt}} - ec_{t,c}^{\text{curt}} < D_{\max}
\end{aligned}
\tag{3}
$$

$\alpha n$ and $\beta n$ are parameters varying from customer to customer. The former is the response to a customer's consumable energy reduction, where a higher value denotes that a customer is likely to become more dissatisfied if the electricity prices become lower. The latter parameter is a predetermined constant. $D_{\min}$ and $D_{\max}$ are the lowest and highest energy reduction, respectively.

Therefore, the minimized cost of a CU n can be described as follows:

$$\min \sum_{t=1}^{T} \left[ \lambda_{t,c} \cdot \left( ec_{t,c}^{\text{curt}} + ec_{t,c}^{\text{critic}} \right) + \varphi_{t,c} \right] \tag{4}$$

where both the cost of a customer n to buy electricity and the dissatisfaction efficient from demand reduction are used.

### 4.1.2. Service Provider Model

Buying electricity at wholesale prices from the GO and later selling that electricity at a retail price to CUs, SPs need to ensure maximum profit. The dynamic pricing can be denoted as follows:

$$\max \sum_{c=1}^{N} \sum_{t=1}^{T} (\lambda_{t,c} - \pi_t) \cdot \left( ec_{t,c}^{\text{curt}} + ec_{t,c}^{\text{critic}} \right)$$
$$\kappa_1 \pi_{t,\min} \leqslant \lambda_{t,n} \leqslant x_2 \pi_{t,\max} \tag{5}$$

$\kappa_1$ and $\kappa_2$ are predetermined. These are the parameters that keep the price fair for both the SP and the CUs, and they are the retail price bound coefficients.

### 4.1.3. Objective Function

Both the SP and CU benefit functions can be denoted as follows:

$$\max \sum_{c=1}^{C} \sum_{t=1}^{T} [\rho \cdot (\lambda_{t,c} - \pi_t) \cdot ec_{t,c} - (1 - \rho) \cdot (\lambda_{t,c} \cdot ec_{t,c} + \varphi_{t,c})]$$
$$ec_{t,c} = ec_{t,c}^{\text{curt}} + ec_{t,c}^{\text{critic}} \tag{6}$$

### 4.2. *Q-Learning-Based Electricity Price Forecasting*

This paper features a hierarchical electricity market consisting of the grid operator (GO), service provider (SP) and customer (CU). The service provider monitors the energy demand and dissatisfaction level of customers. Monitored data are then reflected by adjusting the dynamic pricing strategy. Wholesale electricity prices provided by grid operators are also monitored for energy consumption. In this system, the service provider works as an agent who determines a retail price to serve the customers. Here, customers serve as the environment. Time is segmented on an hourly basis, and each time slot reward is generated from customers as an electric bill. Customer's energy demand and dissatisfactory factors will influence the state of dynamic pricing, and it will be determined with the help of *Q*-learning.

Here, the implementation of *Q*-learning can provide significant advantages. As RL is model-free, it enables the determination of price actions without any model environment. The trial-and-error process of RL comes into play as customers and service providers dynamically set up the price and profit. Secondly, the adaptability of *Q*-learning is a key factor. The electricity market is changing massively day by day with the demand–supply, price factor and other factors such as customer dissatisfaction. *Q*-learning is adaptive to coping with the changes through its ongoing learning process. Thus, the flexibility of the dynamic energy market is kept in check.

The proposed system will support a pricing strategy that will be applied in a hierarchical electricity market. The hierarchical framework of this pricing is formulated with RL as a Markov decision process (MDP). Finally, this decision-making of dynamic retail pricing is solved with *Q*-learning. Being model-free, the system will learn gradually about the flexibility and uncertainties of the demand change and system requirements over time through the online learning process. Customer dissatisfaction levels are also taken as a key factor in changing customers' demand usage through the online learning process.

As previously mentioned, the proposed system can be structured in an RL method where service providers will serve as an agent, customers will serve as the environment, the action will be the retail price that the customers are provided by the service providers, the state will be represented by the energy demand, consumption and dissatisfaction levels,

and finally, both the SPs' profit as well as CUs' minimized cost can be seen as a reward. Here, the Markov decision process (MDP) is used to formulate retail pricing, and then, using *Q*-learning, the dynamic pricing algorithm is formulated.

### 4.2.1. Producer Input Selection

The system uses a 0/1 knapsack to determine the best possible value for electricity produced from an array of producers in the market. Knapsack derives a case in which the aim is to maximize the value of a knapsack while staying within the weight limitation. As per the functionality of a 0/1 knapsack, it considers either an entire item or rejects it altogether, given a collection of objects with certain weights and corresponding weight values. The algorithm takes the electricity produced per unit from the produced as weight and its corresponding wholesale price as the value of electricity. Moreover, the highest demand from customers is taken as the maximum weight capacity of the algorithm. Therefore, the maximum customer demand W is the knapsack capacity. Here, the two arrays, wholesalePrice [0. . . n − 1] and producerGenerator [0. . . n − 1], represent the wholesale price of electricity and the amount of electricity produced from each producer in a certain region. The system uses a 0/1 knapsack since electricity units cannot be broken down and a 0/1 knapsack takes complete units.

### 4.2.2. Formulating System Model to Markov Decision Process

As the dynamic electricity market has a stochastic environment, MDP is beneficial for reforming the system model. Only the current time slot will be considered for reward and energy consumption, and therefore, no historical data will have any impact on maintaining the stochastic feature of the environment. The MDP includes these major four components:

(1) $t$ defines the time interval for the actions that represent retail price. It has to be discrete.
(2) $\lambda_{t,c}$ is the retail price chosen at time $t$ for CU $c$.
(3) $ed_{t,c}$ represents a CUs energy demand before being notified of the retail price from SP. $ec_t$, is the consumption that occurs after the price signal.
(4) $r(ec_{t,c}|ed_{t,c}, \lambda_{t,c})$ is the reward that defines a minimal cost of CU $c$ and SP's maximum profit at time $t$.

Thus, for one episode, the reward will be

$$R = r(ec_{1,n}|ed_{1,c}, \lambda_{1,c}) + r(ec_{2,c}|ed_{2,n}, \lambda_{2,c}) + \cdots + r(ec_{T,c}|ed_{T,c}, \lambda_{T,c})r(ec_{t,c}|ed_{t,c}, \lambda_{t,c})$$
$$= \sum_{c=1}^{C}[\rho \cdot (\lambda_{t,c} - \pi_t) \cdot ec_{t,c} - (1 - \rho) \cdot (\lambda_{t,c} \cdot ec_{t,c} + \varphi_{t,c})] \tag{7}$$

The total future reward will be

$$R_t = r(ec_{t,c}|ed_{t,c}, \lambda_{t,c}) + r(ec_{t+1,c}|ed_{t+1,c}, \lambda_{t+1,c}) + \cdots + r(ec_{T,c}|ed_{T,c}, \lambda_{T,c}) \tag{8}$$

As the environment is stochastic, the rewards for the same actions can also diverge significantly. Therefore, a discounted future reward is used.

$$R_t = r(ec_{t,c}|ed_{t,c}\lambda_{t,c}) + \gamma \cdot r(ec_{t+1,c}|ed_{t+1,c}, \lambda_{t+1,c}) + \gamma^2 \cdot r(ec_{t+2,c}|ed_{t+2,c}|\lambda_{t+2,c}) + \cdots + \gamma^{T-t} \cdot r(ec_{T,c}|ed_{T,c}, \lambda_{T,c}) \tag{9}$$

where $\lambda \in [0, 1]$ is a discount factor that compares future reward with current reward system. A value of 1 for $\gamma$ means that the same action implemented on the environment will result in the same reward each time, resulting in a deterministic environment. The pricing strategy that maps current states to action will be $v : \lambda_{t,n} = v(ed_{t,c})$. With the help of *Q*-learning, the optimal policy $v$ will be determined to maximize reward.

### 4.2.3. Using *Q*-Learning for Dynamic Pricing Problem

*Q*-learning is a subsection of RL that is model-free. It can be used to obtain the optimal policy, which is the dynamic policy referred to in this paper. The *Q*-learning algorithm is as follows:

The main $Q$-learning process involves a $Q$-value $Q(ec_{t,c}|ed_{t,c}, \lambda_{t,c})$, which is assigned to every state-action pair at a time slot $t$. Then, it is updated over each episode that promotes good behavior. $Q^*(ec_{t,c}|ed_{t,c}, \lambda_{t,c})$ refers to the maximum discounted reward for the future when taking the $\gamma_{t,c}$ action. This optimal $Q$ value can be referred to below:

$$Q^*(ec_{t,c}|ed_{t,c}, \lambda_{t,c}) = r(ec_{t,c}|ed_{t,c}, \lambda_{t,c}) \quad + \gamma \cdot \max Q(ec_{t+1,c}|ed_{t+1,c}, \lambda_{t+1,c}) \qquad (10)$$

According to the algorithm, the maximum $Q$-value is calculated. If the learning rate factor is 0, then the agent learns nothing from the existing data, and in this case, it is the SP who gains no new information regarding the optimal policy. On the other hand, a learning factor of 1 will make the agent reevaluate the recent state of the environment; in this case, it is considering the recent demand response of the customers.

In the dynamic electricity market, the service provider interacts with the customer through their dynamic pricing. Then, the demand for CUs change overtime and the SPs receive a new state. The trial and error of these set of actions, which is the dynamic pricing implemented by the SPs, generates $Q$-values that are stored and updated from time to time. Eventually, the values are going to converge to a maximum value. If the maximum expected profit of SPs with action $\lambda_{t,c}$ is at a demand $ed_{t,c}$, the optimal policy $v$ can be referred as:

$$C = \operatorname{argmax} Q(ec_{t_u D}|ed_{t_t,0}, \lambda_{t,0}) \qquad (11)$$

Figure 4 shows the flowchart of how the algorithm works:

Since RL excels at making sequential decisions in an unknown environment, it can adapt to the policy that is required in real-time and learns from past experiences. Therefore, here, $Q$-learning is the best RL-method to find the optimal pricing strategy. Here, the flowchart shows that the algorithm begins at 00:00 and ends after 24 h. The inputs that are taken are the prices from producers following the time slot T in an hourly fashion. There are also the coefficients of the price bounds from the third-party providers and all other parameters in the flowchart shown above. After the algorithm runs the inputs, it initializes the $Q$-value $Q(ec|ed)$ to 0, the time to the beginning of the day. The algorithm then finds the optimal prices at each hour of the day using the epsilon-greedy policy, abiding by the price bounds. To be efficient, the epsilon-greedy policy selects an action with uniform distribution from a set of available actions.

Using this policy, a random action with a probability $\epsilon$ that is between 0 and 1 from the 1-values from each state is performed during the iteration. In short, the agent here randomly selects a retail price at each state and scans them amongst the already stored values from the previous states to select the highest value and then choose the retail price.

After the SP chooses the retail price, it will receive rewards accordingly from the reward equation. At the same time, the SP also observes the customer's demands for the following time slot and updates the $Q$-values using the $Q$-learning process. Lastly, if the $Q$-value does not converge to the maximum $Q$-value, the system then moves on to the next iteration until it finally does converge. This is the termination rule, where the difference between the present and previous $Q$-value is less than $\delta$. After converging, the SP will then obtain the final responsive retail prices for the hourly slots of the day.

The proposed $Q$-learning Algorithm 1 is given below:

**Algorithm 1:** Proposed *Q*-learning algorithm

S = State, A = Action

*Initialize*

Initialize Q(S, A) using arbitrary values

**for** *each iteration i*, **do**

At time interval *t* Choose an action $\lambda_{t,c}$ and execute for state $(ed_{t,c})$

Observe the reward $r(ec_{t,c}|ed_{t,c}, \lambda_{t,c})$ and the new state $(ed_{t+1,c})$

$Q(ec_{t,c}|ed_{t,c}, \lambda_{t,c}) \leftarrow Q(ec_{t,c}|ed_{t,c}\lambda_{t,c}) + \theta \cdot [r(ec_{t,c}|ed_{t,c}\lambda_{t,c}) +$
$$\gamma \cdot \max Q(ed_{t+1,c}|ed_{t+1,c}, \lambda_{t+1,c}) - Q(ed_{t,c}|ed_{t,c}\lambda_{t,c})]$$

**end**

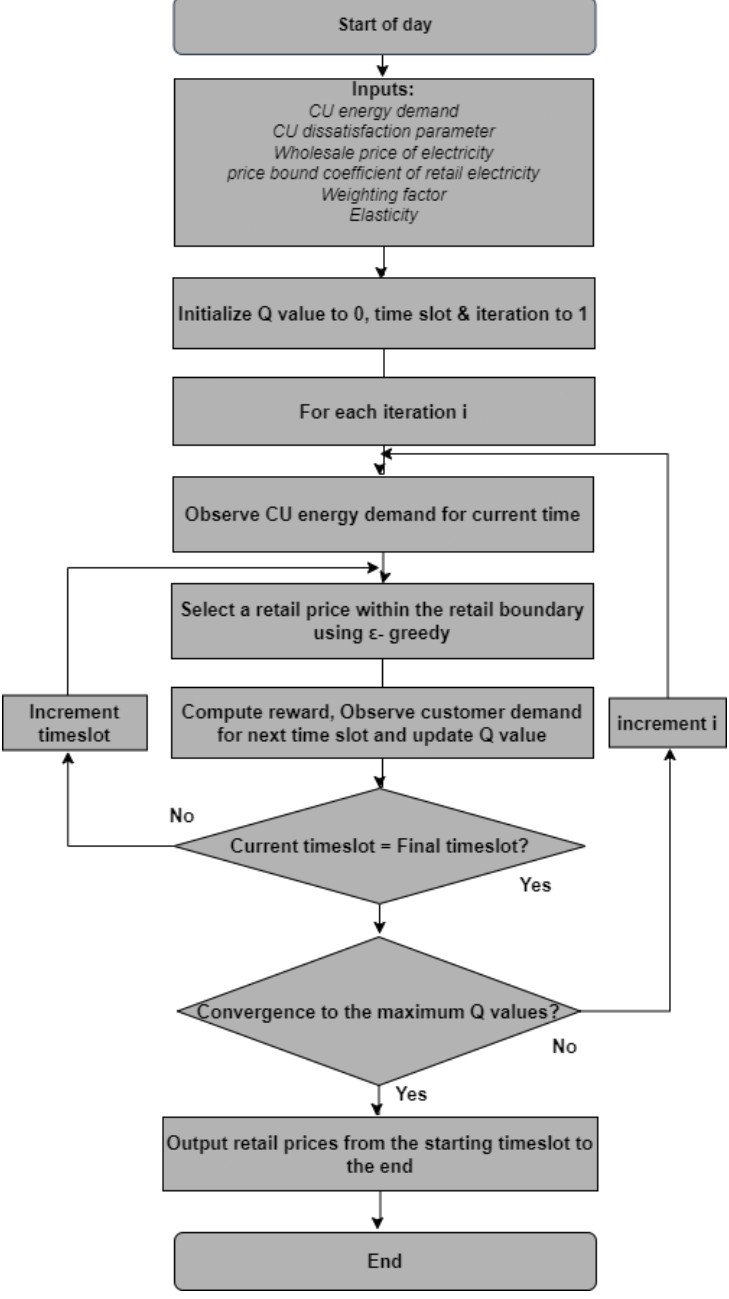

**Figure 4.** Flowchart for implementing the *Q*-learning process for figuring out the optimal price.

## 5. Implementation and Analysis

### 5.1. Implementation of Blockchain

In this section, an Ethereum-based private blockchain system is introduced that ensures a safe and secure electricity marketplace between the CU and SP. Ethereum provides an open-source blockchain platform that runs based on smart contracts, and the currency it runs with is 'Ether'. It is mainly a DPL that can record, verify, and deploy transactions that occur within the network. A 'DApp' [26] or decentralized application is introduced to complete the transactions, and the application users pay a particular fee that is 'gas' to run the functions, which depends on the amount of computational power needed to compute the whole process. It works on the 'Proof of Work' consensus mechanism. There is a canonical computer in the Ethereum technology (EVM) whose state is followed by the other nodes. The main smart contract is stored in the EVM, and users call out programs into the EVM storage using particular parameters to perform actions. In this implementation, a complete business DApp is demonstrated. In this system, the smart contract was created and deployed in the blockchain system as well as the front end, and the transactions were confirmed and stored in the updated blockchain network. Here, we present the work plan for the implementation system. The basic components of the implemented system and the way they intercommunicate are shown in Figure 5.

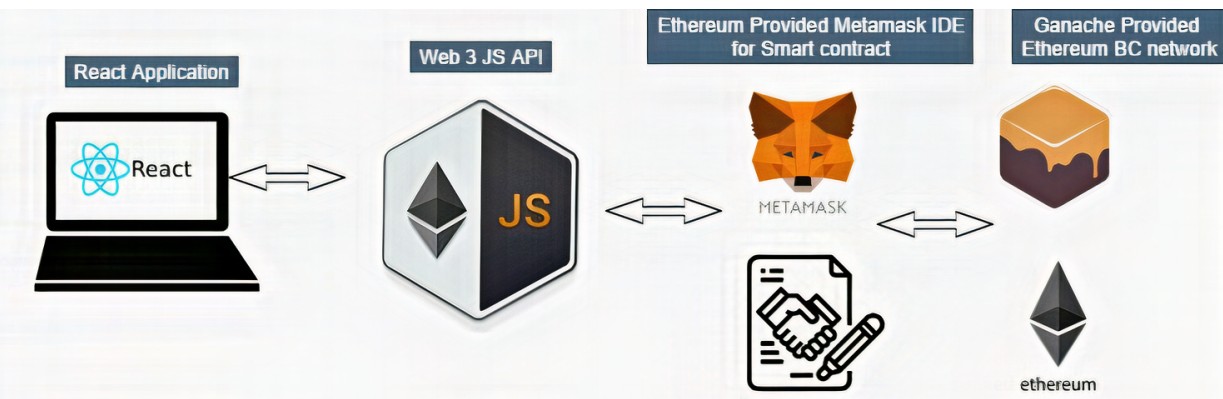

**Figure 5.** Workplan of the blockchain implementation.

### 5.2. Experimental Setup

#### 5.2.1. Blockchain Network

The core network of the system is Ganache, which is a personal or private blockchain for DApp development [27]. Its primary purpose is to design, launch, and test the DApp securely and easily without costing any real Ether. It provides a local dummy server that holds ten (10) accounts containing a balance of 100 fake Ether to perform the transactions. Each account holds a public key and a private key that is used to conduct the transactions. The server only accepts the RPC connection on host http://127.0.0.1:7545 (accessed on 24 January 2022). In Figure 6, 10 fake accounts are displayed, where each has a balance of 100 Ether. Using the private key, the system can connect that particular account with the browser to perform transactions. It shows the visual interface for Ganache CLI, where functions and features are displayed to the users.

**Figure 6.** Ganache UI blockchain server.

### 5.2.2. Smart Contract Deploy

The system uses Solidity language to write the smart contract, and the version of the Solidity language is >=0.4.21 <0.6.0. It holds the main functions and events of the whole system. As the smart contract contains the main structure of the performed actions, it is necessary to provide accurate and realistic parameters to meet the particular requirements. After deploying the smart contract, it will hold an address of its own that must be provided to the blockchain while connecting them. In our implementation, a smart contract named 'Marketplace.sol' is initiated that holds the records of the id, name, price, owner's address, and purchase history. It also consists of several events that can run programs and store variables. The functions are createElectricity (name,price) and purchaseElectricity (id). These functions need accurate and correct values in the parameters in order to run. After that, the events are executed.

### 5.2.3. Web3.js

Web3.js is a set of libraries that allows a system to communicate with an Ethereum node, either locally or remotely, using an HTTP or IPC connection. It establishes a connection between the smart contract and the blockchain. The Ethereum blockchain is accessed with the web3 JavaScript library. It can retrieve user accounts, send transactions, and communicate with smart contracts among other things. Several utility functions are also provided by using the web3.js, which makes development easier. Furthermore, the smart contract is loaded into the blockchain using web3.js. It is an API that connects every component of the whole application.

### 5.2.4. MetaMask Wallet

MetaMask Wallet is a cryptocurrency wallet that permits users to communicate with a blockchain with the help of a browser extension. The wallets provided by Ganache must be imported into the MetaMask Wallet using the private key of the accounts. A custom RPC must be created, which holds the IP http://127.0.0.1:7545 (accessed on 24 January 2022), the same as the Ganache RPC. These accounts confirm transactions and pay for the gas price required for buying energy. One of the imported accounts is named 'Seller', which can add selling items to the list, and another account is named 'Buyer', which can buy the listed

items and be the owner of that particular product. After that, they perform transactions. The user interface of the MetaMask wallet browser extension is displayed in Figure 7.

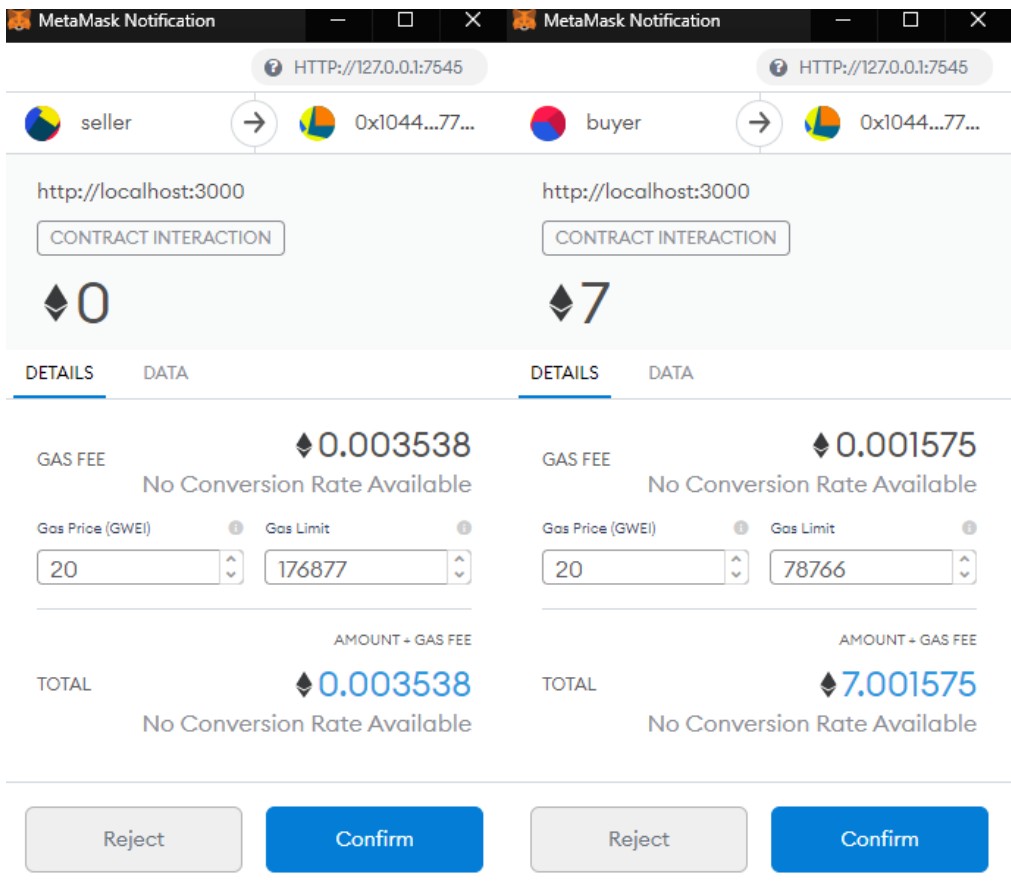

**Figure 7.** MetaMask wallet for seller and buyer.

Table 1 contains the information on created blocks in the blockchain. It also stores the timestamps of the mined blocks and the amount of gas used to conduct that transaction.

**Table 1.** Gas used for every block.

| Block No | Mined Date | Gas Used |
|----------|------------|----------|
| 82 | 2 June 2021 3:53:33 | 52,511 |
| 81 | 2 June 2021 3:53:01 | 117,930 |
| 80 | 2 June 2021 3:49:43 | 117,954 |
| 79 | 2 June 2021 3:48:47 | 117,918 |
| 78 | 2 June 2021 3:48:17 | 117,906 |
| 77 | 2 June 2021 3:48:02 | 132,894 |
| 76 | 2 June 2021 3:45:42 | 745,906 |
| 75 | 2 June 2021 3:45:42 | 244,636 |

### 5.2.5. React.js

The front end of the application is made with React.js. React is a front-end JavaScript library for creating user interfaces and UI designs. The front end connects to the blockchain using the web3 functions. Every time the seller adds an item to the list by providing the parameters (name, price), an item is added to the' Buy Electricity' list. It will also show the owner's address of the item, which will be the seller while the items are listed. Then the buyer can buy those particular items by pressing the' Buy' button and confirming the transactions with MetaMask Wallet. As a result, the Ether will be transferred to the seller

from the buyer, and the ownership of the product will be swapped, where the buyer's address will then be presented.

Figure 8 depicts the user-friendly interface of the web application where buyers and sellers can make transactions.

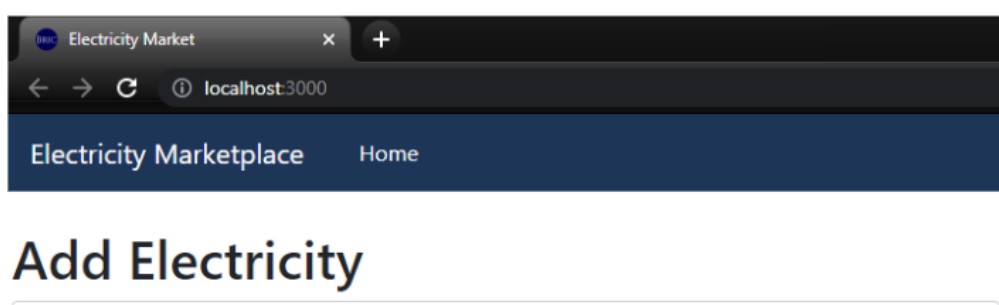

**Figure 8.** Front end application using React.js.

*5.3. Implementation of* Q-*Learning-Based Electricity Price Forecasting*

Input Data

- **Producer Input**  Part of the input has five different electricity producers in the form of coal, nuclear, wind, water, and air. Since each of the producers produces different quantities of electricity at different times of the day and prices them accordingly, the simulation uses a 0/1 knapsack as the algorithm to choose the best producer with the best price for retailers. The algorithm takes the maximum capacity of weight as list W for each hour of the day [0,1,2,3], the list that contains the electricity production weight in list wt [C1, C2, C3, . . .], and the prices for production in values V [V1, V2, V3, V4]. The algorithm filters through the list through brute force recursion, and it calculates the total weight and value of all the subsets. Moreover, it will only consider the subsets whose total weight is smaller than the maximum capacity W. The values V are shown in Figure 9 and they show the difference in prices amongst four competitive producers. Moreover, the figure also compares the producer prices with the wholesale price.

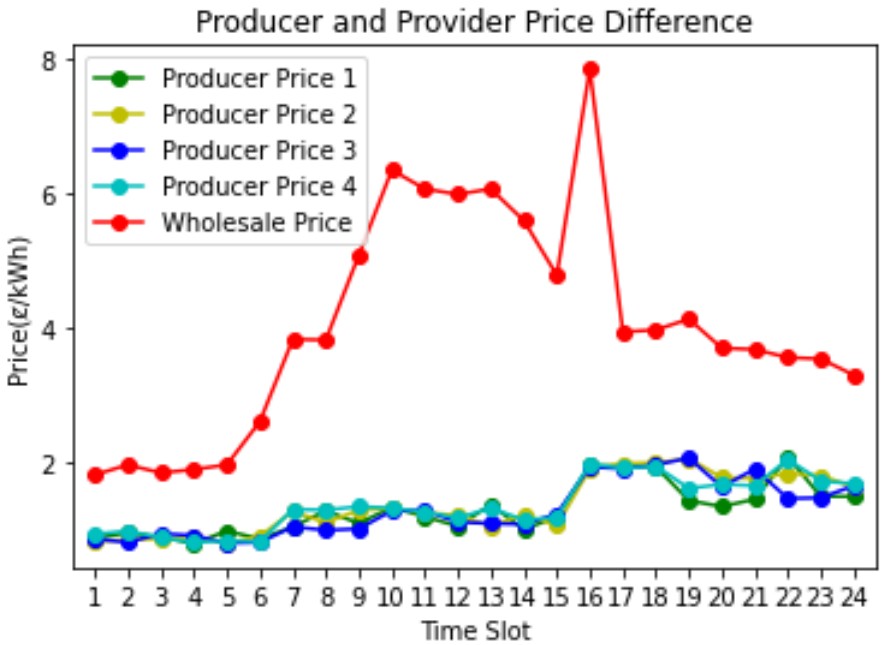

**Figure 9.** Different prices of producer and provider.

- **Customer Input** The other inputs that the algorithm takes are the dissatisfaction parameters—dmul, alphan, and betan. Moreover, the customers' curtailable demand and critical demand shown in Figures 10–12 are also used for determining the optimal retail price at specific times of the day, given the high demand during peak hours and the low price at off-peak hours.

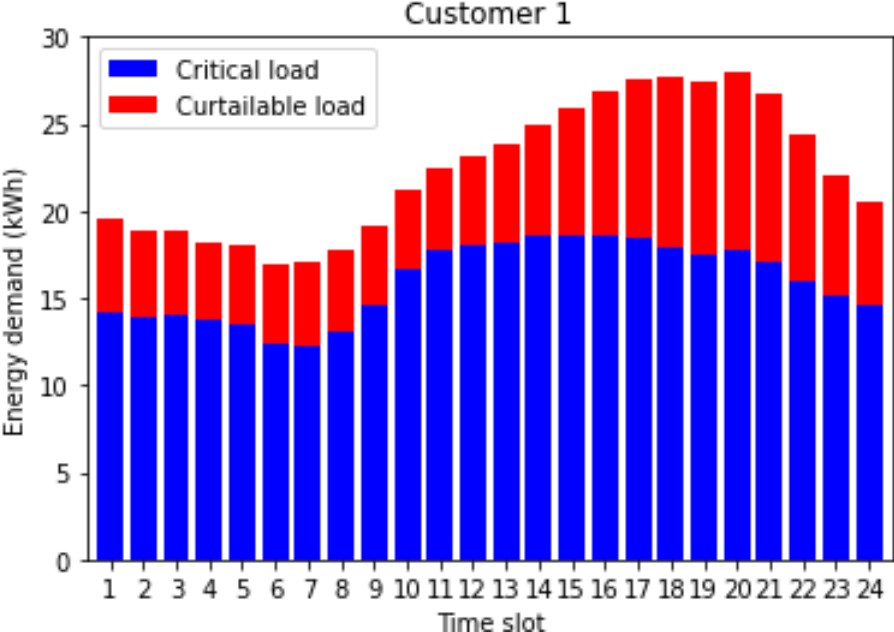

**Figure 10.** Critical and curtailable load for customer 1.

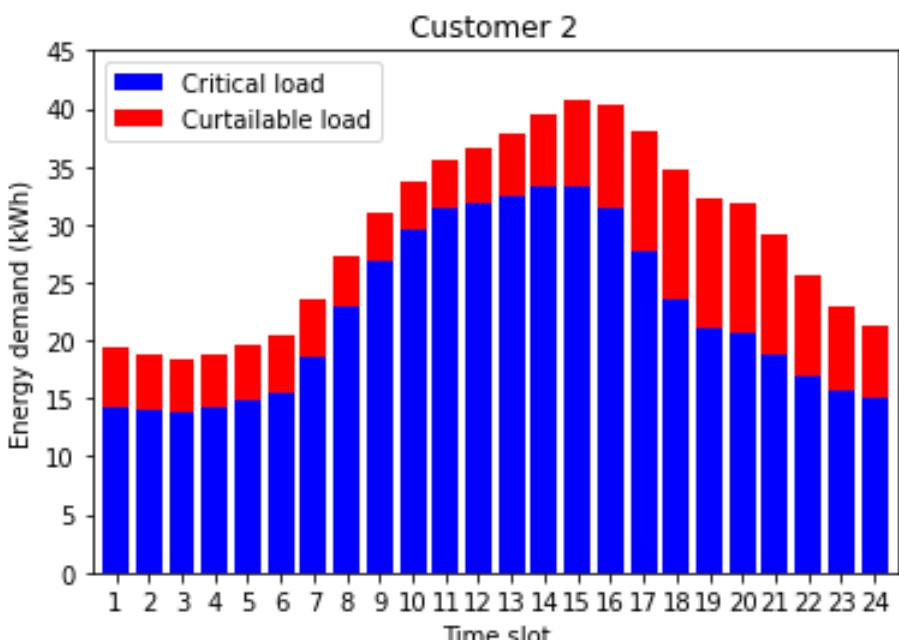

**Figure 11.** Critical and curtailable load for customer 2.

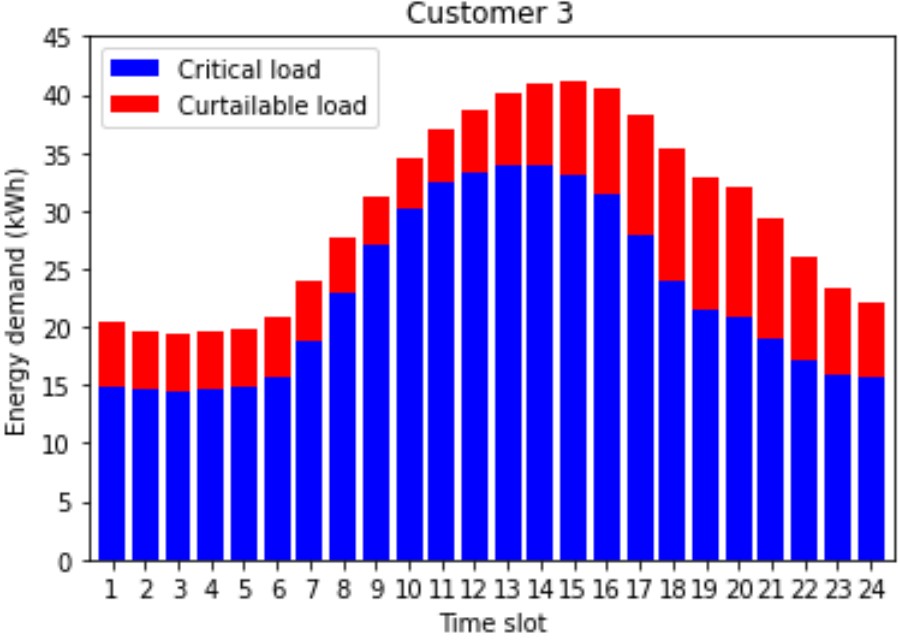

**Figure 12.** Critical and curtailable load for customer 3.

*5.4. Numerical Simulation Results*

Figures 13–15 show the simulations of the energy demand, consumption, the wholesale price, and the retail price per hour for an entire day. Each of the time slots represents the value of T in the sections above. Here, all the parameter values are specific, and they can be changed as per the characteristics of the market and the providers, producers, and the customers. However, they do not change the overall performance of the simulation in a negative way.

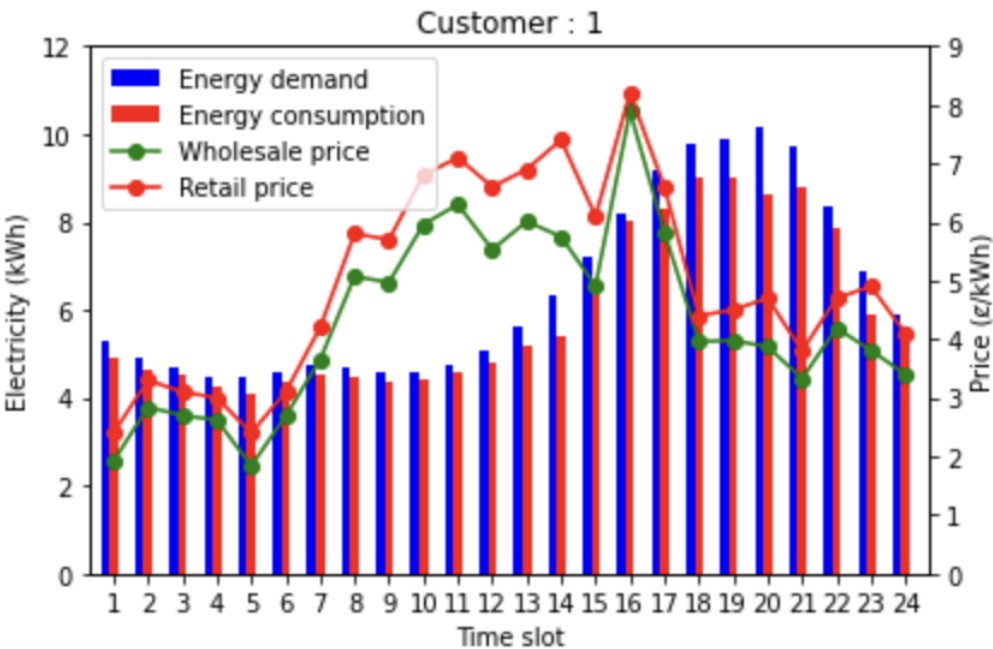

**Figure 13.** Energy demand and consumption with optimal retail price for customer 1.

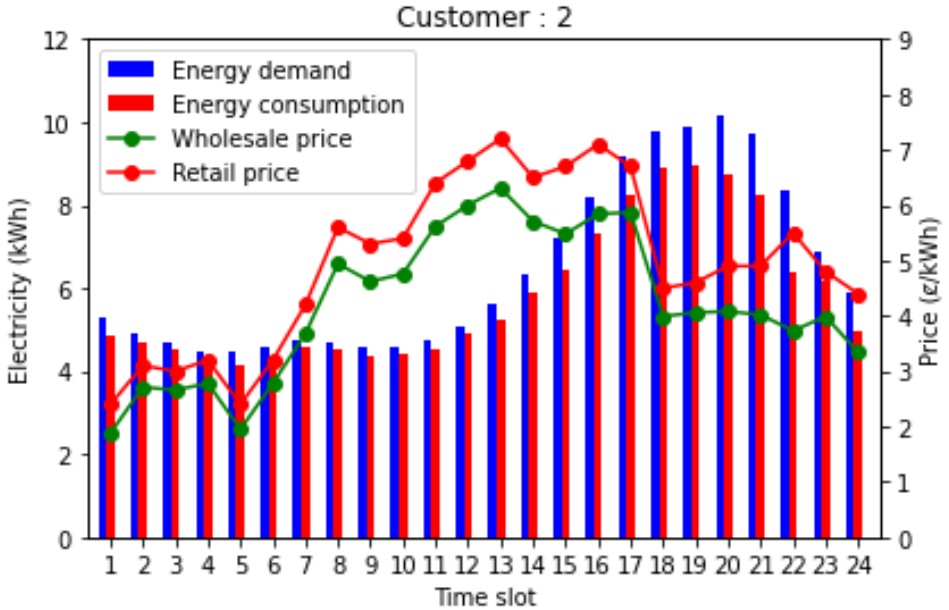

**Figure 14.** Energy demand and consumption with optimal retail price for Customer 2.

From the graphs in Figures 16–18, it can be seen that the agent at the beginning does not know the best actions that result in the higher *Q*-values; however, as the algorithm iterates, the agent learns from the environment gradually and then finally converges to the maximum *Q*-value.

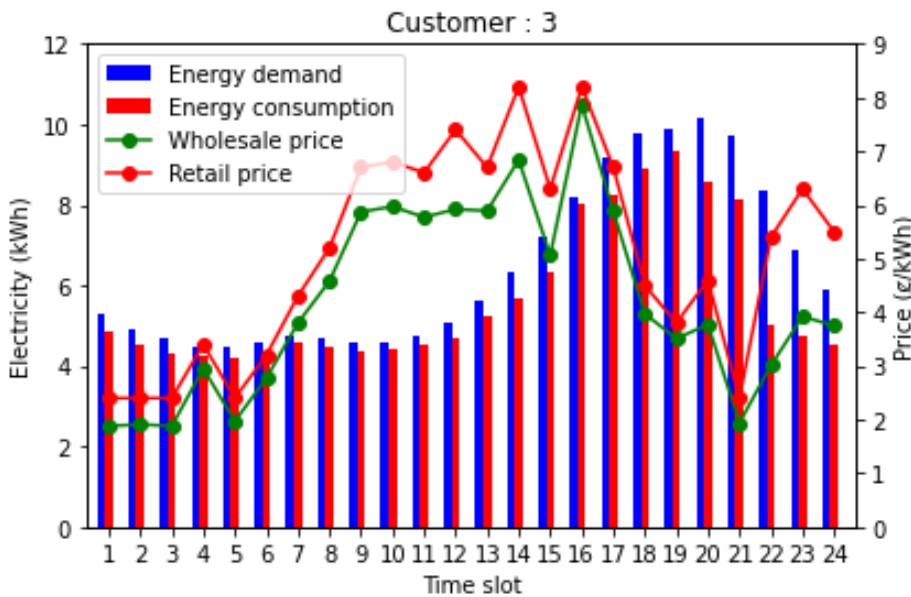

**Figure 15.** Energy demand and consumption with optimal retail price for customer 3.

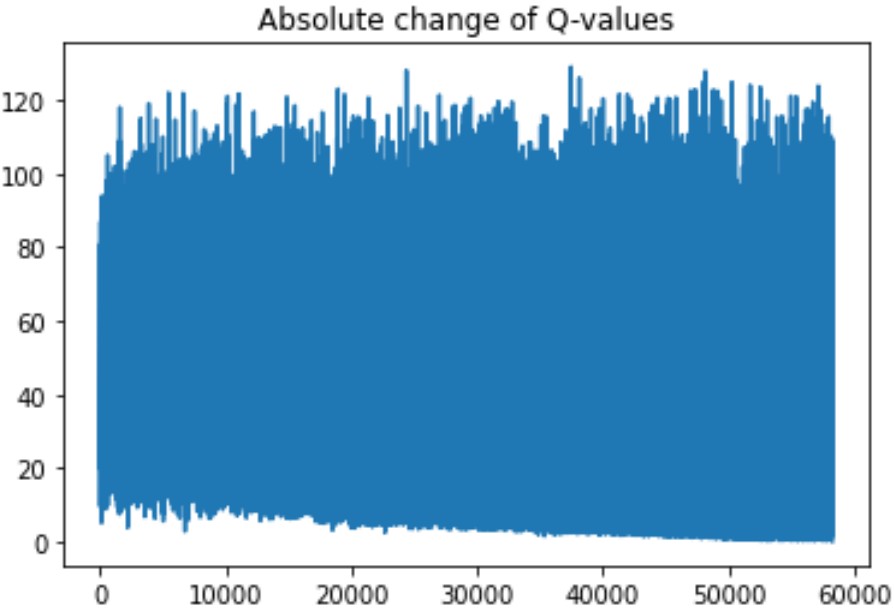

**Figure 16.** Change of *Q* values.

Optimal Retail Prices

The simulation results in the optimal retail prices for each of the customers in question. Each of the figures shows the optimal retail price along with the prices from the providers for each time slot T. Moreover, the curtailable load and energy consumption is also shown for the customers. The graph also shows that the retail price shows a very similar pattern to the wholesale price but never exceeds it due to the price bounds and dissatisfaction parameters. The price elasticity of the entire day shows a continuing increase in demand as well as the increase in retail price and then a gradual decrease in both. This is because, during peak hours, the electricity demand is maximum and more elastic; therefore, it results in a higher energy consumption.

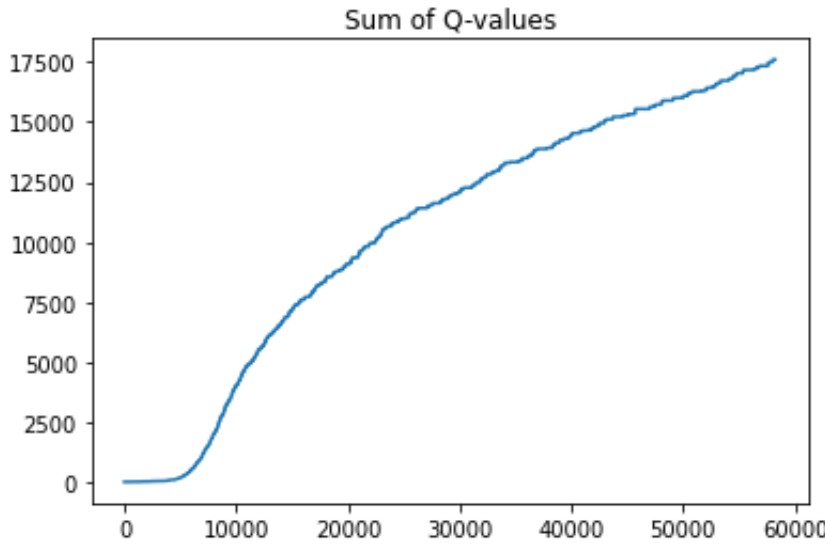

**Figure 17.** Sum of *Q* values.

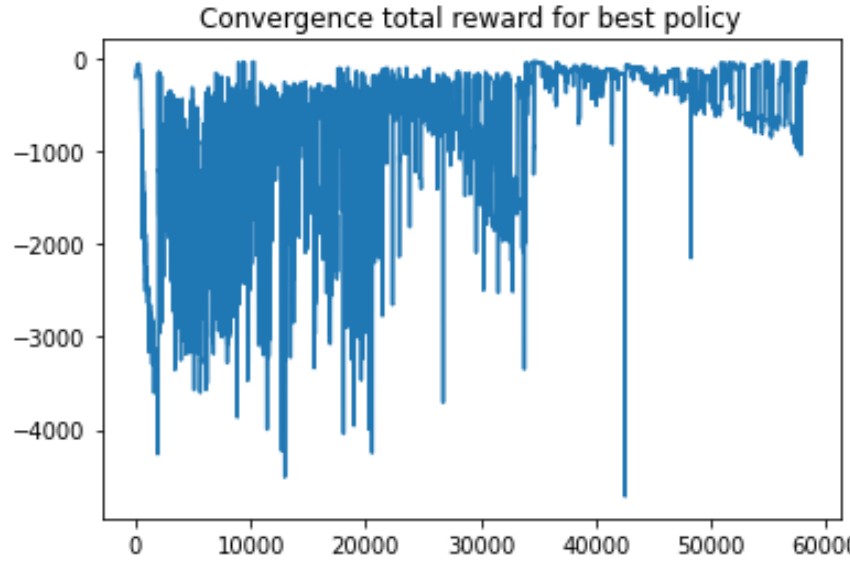

**Figure 18.** Convergence total reward for best policy.

## 6. Conclusions and Future Works

In this paper, the major drawbacks of traditional grids as well as conventional electricity pricing strategy implementations, which were based on abstract models, are discussed. This paper proposes a dynamic DP algorithm, which will not only benefit the SP profit but also minimize CU costs. This model-free approach was performed using RL, where retail prices will be adaptive in nature and change based on the investigation of CUs' and dissatisfaction level and demand profile. This dynamic pricing problem of electricity is approached by converting it to a finite discrete MDP and then formulating the decision-making using *Q*-learning. As a result, an SP does not need any specific model of the customers' energy consumption to learn about future retail rates. Rather, it is more convenient as the SPs will learn about CUs' satisfaction and dissatisfaction levels through dynamic online interaction. This approach is feasible as, with the change of environment, which in this case is the variation of CUs and their demand profile, the dynamic pricing model will improvise through learning. Thus, it is beneficial for both the SP and CU, ensuring a win–win strategy. Furthermore, a secured blockchain transaction is paramount for the security of both the SP and CU, which is also discussed in this paper. In this paper, a blockchain based transaction

DApp has been created where it works as a virtual electricity market place for CU. SPs will list their selected prices for a particular amount of load in the virtual market, and CUs can buy it using Ether. As the transaction is decentralized and performed in cryptocurrency, the maximum security of the user's transaction activity is secured. The blockchain environment is created using a personal blockchain named Ganache. The frontend is constructed using react.js and the backend with smart contracts, which are a cumulation of transaction functions. Web3.js is used to connect the backend and front end. Metamask works as an online wallet to connect the blockchain with a browser.

In the future, the current dynamic pricing model will be further improved upon by deeply analyzing the weighting factor and further scrutinizing the optimal between the provider, producer and customers. The blockchain system can be improved by implementing advanced storage options such as the interplanetary file sharing system (IPFS), which acts as decentralized online storage to store the information of users. User data in IPFS can have an encryption facility for more security and convenience. Although the blockchain transaction is highly secured, the PoW needed for authentication is relatively energy consuming. It takes a lot of computational power; therefore, in cases where a fast transaction is needed, efficiency can sometimes be an issue. This efficiency can be highly negligible in most cases, given the secured, reliable transaction method the blockchain system provides. Therefore, even though the computational power required for such a system is very high, the trade-off here is that security is heavily enhanced as a result.

**Author Contributions:** Conceptualization, M.M.M.A.M., R.S.U. and M.G.R.A.; methodology, M.M.M.A.M., R.S.U., M.A.H., T.B.S., M.G.R.A., M.M.H. and M.R.H.; software, M.A.H., T.B.S.; validation, M.M.M.A.M., R.S.U., M.A.H. and M.G.R.A.; formal analysis, R.S.U., M.M.H. and M.R.H.; investigation, M.M.M.A.M., R.S.U., M.A.H., T.B.S., M.G.R.A. and M.M.H.; resources, M.M.M.A.M.; data curation, M.M.M.A.M., R.S.U., M.A.H., T.B.S., M.G.R.A. and M.M.H.; writing—original draft preparation, M.M.M.A.M., R.S.U., M.A.H., T.B.S., M.G.R.A. and M.M.H.; writing—review and editing, M.M.M.A.M., R.S.U., M.A.H., T.B.S., M.G.R.A., M.M.H. and M.R.H.; supervision, M.G.R.A.; funding acquisition, M.M.H. All authors have read and agreed to the published version of the manuscript.

**Funding:** This work was supported by King Saud University, Riyadh, Saudi Arabia, through Researchers Supporting Project number RSP-2021/18. This work was also supported in part by the MEIF-SCI FY2021 grant of the University of Maine System.

**Institutional Review Board Statement:** Not applicable.

**Informed Consent Statement:** Not applicable.

**Data Availability Statement:** Not applicable.

**Conflicts of Interest:** The authors declare no conflict of interest.

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
