# Peer review of "Blockchain Based Smart-Grid Stackelberg Model for Electricity Trading and Price Forecasting Using Reinforcement Learning"

_applsci, doi:10.3390/app12105144_

Round 1

Reviewer 1 Report

The manuscript looks more like a report than a scientific article. The novelty of the manuscript is not clear. It can not be published in this form.   

Author Response

Reviewer 1, General Comments: The manuscript looks more like a report than a scientific article. The novelty of the manuscript is not clear. It can not be published in this form.   

Author response: Thanks to the respected reviewer to review our manuscript. We have thoroughly reviewed and updated our manuscript for improving the readability and structure of this research article. As per the reviewer’s concern, we have pointed out the novel contributions of this research in the introduction section of the revised manuscript. We hope the revised manuscript will meet the requirements of a scientific article.

Author Action:

[Updated][Abstract: Page 1]

Abstract: A smart grid is an intelligent electricity network that allows efficient electricity distribution from source to consumers through telecommunication technology. The legacy smart grid follows the centralized oligopoly marketplace for electricity trading. This research proposes a Blockchain-based electricity marketplace for the smart grid environment to introduce decentralized Leger in electricity market for enabling trust and traceability among the stakeholders. The electricity prices in smart grid are dynamic in nature. Therefore, the price forecasting in smart grids has paramount importance for the service providers to ensure service level agreement and also to maximize profit. This research introduced a Stackelberg Model based dynamic retail price forecasting of electricity in smart grid. The Stackelberg model considered two stage pricing between electricity producers to retailers, and retailers to customers. To enable adaptive and dynamic price forecasting the reinforcement learning is used. Reinforcement Learning provides an optimal price forecasting strategy through on-line learning process of the system. Moreover, reinforcement learning will provide model-free approach which will make the system smarter for understanding customers’ energy usage pattern. The use of blockchain will connect the service providers and consumers in a more secure transaction environment. It will help tackling the centralized system’s vulnerability by performing transactions through customers’ smart contracts. Thus, the integration of blockchain will not only make the smart grid system more secure, but also price forecasting with reinforcement learning will make it more optimized and scalable.

[Updated][Section 1: Introduction, Paragraph: 4, Page 2]

Therefore, the key contributions of this research are as follows:

  • This research introduced a Stackelberg Model based dynamic retail price forecasting of electricity in smart grid. The Stackelberg model considered two stage pricing between electricity producers to retailers, and retailers to customers. To enable adaptive and dynamic price forecasting the reinforcement learning is used.
  • A Blockchain-based electricity marketplace is proposed for the smart grid environ- ment for enabling decentralized Leger in electricity market.
  • The Blockchain-based smart grid electricity marketplace is implemented and the sim- ulation of the system returns responsive retail prices, change in energy consumption due to change in price, and the price pattern for an entire day. Moreover, its asses the quality and performance of the dynamic pricing system for the demand response program. The simulation of an entire day for each customers show that retail price never falls below the wholesale price; however, it also changes to a price as close to the wholesale price when dissatisfaction of customers is at maximum due to a rise in demand at lower consumption rate. Therefore, the simulation shows that the prices are responsive for both retailers and customers.

Reviewer 1, Concern # 1:  Does the introduction provide sufficient background and include all relevant references?

Author response: We would like to thank to the honorable reviewer for the insightful question. As per reviewer’s instruction, we have improved the introduction my following the instructions of the second reviewers by adding the required necessities. We have added important background studies as well as relevant and important papers and references as per instruction.

[Updated][Section 1: Introduction, Paragraph: 2, Page 2]

Demand response (DR) is an efficient method to use in smart grid system to reduce cost and improve grid efficiency. Price based DR refers to influencing the customers’ electricity usage with the variable electricity prices. To help with dynamic pricing in a hierarchical energy market, this paper proposes a DR algorithm that keeps up with the dynamic pricing while also reducing the service provider’s (SP) and

customer’s (CU) costs. With the help of Reinforcement Learning (RL) and Q-Learning we are predicting the price of electricity. The SPs get the flexibility to set the price dynamically with accordance to demand and level of dissatisfaction. Furthermore, blockchain integration to every customer’s profile will help to secure the decentralized transaction of electricity between SP and CU.

[Updated][Section 1: Introduction, Paragraph: 3, Page 2]

Though the contributions of the papers mentioned above helped the electricity demand response field to a good extent, it still lacks the dynamic market where demand is ever-changing. Thus, a DR strategy with dynamic pricing compatibility is apt for modern usage.

[Updated][Section 1: Introduction, Paragraph: 7, Page 2]

The simulation of the system returns responsive retail prices, change in energy consumption due to change in price, and the price pattern for an entire day. Moreover, its asses the quality and performance of the dynamic pricing system for the demand response program. The simulation of an entire day for each customers show that retail price never falls below the wholesale price; however, it also changes to a price as close to the wholesale price when dissatisfaction of customers is at maximum due to a rise in demand at lower consumption rate. Therefore, the simulation shows that the prices are responsive for both retailers and customers.

Reviewer 1, Concern # 2: Are the conclusions supported by the results?

Author response: We thank the honorable reviewer for the astute question. Along with other updates from the second honorable reviewer, have updated the conclusion to support the results better and explained why the algorithm, Q-learning mechanism, and the price forecasting comes together, and why they support the results and as such, the conclusion.

Author Action: The whole paper has been put through modifications for rectification of adequate description and explanations of methods and results. Moreover, the conclusion and results have been improve to support each other better.

[Updated][ Section 6: Conclusion and Future Works, Paragraph 2, Page 21]

In the future, the current dynamic pricing model will be further improved upon by analyzing deeply the weighting factor and further scrutinize the optimal between the provider, producer and customers. The blockchain system can be improved by implementing advanced storage options like the Interplanetary File Sharing System (IPFS) which acts as decentralized online storage to store the information of users. User data in IPFS can have encryption facility for more security and convenience. Although the blockchain transaction is highly secured, the PoW needed for authentication is relatively energy consuming. It takes a lot of computational power, so, in cases where fast transaction is needed, efficiency can sometimes be an issue. This efficiency can be highly negligible in most cases given the secured, reliable transaction method the blockchain system provides. Therefore, even though the computational power required for such a system is very high, the trade-off here is that security is heavily enhanced as a result.

Reviewer 2 Report

Major Comments

  • I think it is important that the authors temper some of their stronger claims. This could be an English language issue, but regardless needs to be addressed.
    • For example, “Thus, a DR strategy with dynamic pricing compatibility is impeccable for modern usage” (page 2) needlessly overstates the claim with “impeccable.” Perhaps “apt.”
    • Similarly, many of the adjectives and adverbs should simply be removed. For example, “immensely” should be removed (page 2).
    • Further, “Therefore, the simulation shows that the prices are optimal for both retailers and customers.” It does not show it is optimal, but rather it shows useful responsiveness. No need to overstate the case here.
  • Rewrite the literature review such that it is now referring to “that paper” and “another paper”

Minor Comments

  • Some terms should be defined on first use to broaden the scope of readers that could read this article in this journal. For example, Knapsack
  • I suggest a large amount of English language/grammar review, especially targeted at reducing the length of sentences to enhance clarity. For example, “To solve the dynamic pricing of a hierarchical energy market, this paper would like to propose a DR algorithm that copes up with the dynamic pricing and also helps to reduce the service provider’s (SP) and customer’s (CU) costs” (page 2). This sentence could be rewritten as “To help with dynamic pricing in a hierarchical energy market, this paper proposes a DR algorithm that keeps up with the dynamic pricing while also reducing the service provider’s (SP) and customer’s (CU) costs.”
  • Further, “A smart contract is a bunch of rules and agreements approved by both parties.” Could be “a set of rules…”
  • There are many more examples and I suggest fixing them.

Author Response

Reviewer 2, Concern # 1:  I think it is important that the authors temper some of their stronger claims. This could be an English language issue, but regardless needs to be addressed.  For example, “Thus, a DR strategy with dynamic pricing compatibility is impeccable for modern usage” (page 2) needlessly overstates the claim with “impeccable.” Perhaps “apt.”

Author response: Thank you for the insightful advice. As recommended by the respected reviewer, in the revised manuscript, we have carefully used the english words to state the research outcomes and claims.

The revised manuscript has been proofread by an advanced writing assistant. We hope that the readability of our revised manuscript is improved through proofreading, rectification of grammatical errors and addressing the comments and recommendations of the honorable reviewers. Furthermore, if necessary, we will take a professional proofreading service.

Author action: The whole paper has been put through modifications for rectification of grammatical errors and improved readability. The wordings have been changed so that they justify the claims properly.

In DR efficiency, we can see several works regarding DR models that help to subsidize customer usage by minimizing costs. There is one such example in [ 2–4 ], where electricity consumption of different home appliances was monitored and time-of-use (TOU) pricing helped to minimize customers’ costs. In [ 5-10], we can see the benefit of predetermined next-day electricity prices, and efficient scheduling to help keep the costs of CU in check. Though the contributions of the papers mentioned above helped the electricity demand response field to a good extent, it still lacks the dynamic market where demand is ever-changing. Thus, a DR strategy with dynamic pricing compatibility is apt for modern usage.

Reviewer 2, Concern # 2: Similarly, many of the adjectives and adverbs should simply be removed. For example, “immensely” should be removed (page 2).

Author response: We would like to thank to the honorable reviewer for the suggestion and recommendation. As per the respected reviewer’s recommendation, unnecessary adjectives and adverbs are removed during the proofreading.

Author action: The whole paper has been proofread for rectification of redundant terms and improved readability. The unneeded terms have been removed so that they justify the claims accurately.

[Updated][Section 1: Introduction, Paragraph: 3, Page 2]

Though the contributions of the papers mentioned above helped the electricity demand response field to a good extent, it still lacks the dynamic market where demand is ever-changing. Thus, a DR strategy with dynamic pricing compatibility is apt for modern usage.

Reviewer 2, Concern # 3: Further, “Therefore, the simulation shows that the prices are optimal for both retailers and customers.” It does not show it is optimal, but rather it shows useful responsiveness. No need to overstate the case here.

Author response: We would like to thank the honorable reviewer for the intuitive comment and suggestion. While proofreading we have changed some terms so that an appropriate statement is stated. As per the reviewer’s suggestion, the mentioned sentence has been changed as needed to state proper claims.

Author action: Modifications made to the manuscript in this regard is as follows:

[Updated][Section 1: Introduction, Paragraph: 7, Page 2]

The simulation of the system returns responsive retail prices, change in energy consumption due to change in price, and the price pattern for an entire day. Moreover, its asses the quality and performance of the dynamic pricing system for the demand response program. The simulation of an entire day for each customers show that retail price never falls below the wholesale price; however, it also changes to a price as close to the wholesale price when dissatisfaction of customers is at maximum due to a rise in demand at lower consumption rate. Therefore, the simulation shows that the prices are responsive for both retailers and customers.

Reviewer 2, Concern # 4: Rewrite the literature review such that it is now referring to “that paper” and “another paper”

Author response: We would like to express our deepest gratitude to the honorable reviewer for the insightful comment. As per the respected reviewer’s recommendation, we have rewritten the literature review such that the referring is in the correct format. Moreover, we reviewed more recent relevant works and added them with our previous literature reviews. The corresponding references are added to the References section.

Author action: We have updated section 2: Literature Review and also the References section.

[Updated][ Section 2: Literature Review, Paragraph 2, Page 3]

  1. Lago et al. suggested several price forecasting methods using Deep Learning techniques. It proposes Deep Neural Networks with a hybrid Long Short-Term Memory and Deep Neural Networks structure to significantly improve prediction accuracy. However, that paper is only compared using only a single dataset. Therefore, it is not suitable to use such a paper for real-life experiments and appliances since there are so many factors to look out from[17].

[Updated][ Section 2: Literature Review, Paragraph 4, Page 3]

  1. Jiang et al.[17] proposed a fair transaction packing strategy for permissioned blockchains using IIot systems based on transaction response time. The FAIR-PACK algorithm uses a heuristic and a min-heap technique to divide the data into two subset sum problems for different variables. Extensive comparisons of time and performance complexity are performed to see how the transaction arrival rate, block generation time, block size, and block validity ratio impact FAIR-PACK performance. The analysis revealed that FAIR-PACK delivers more fairness and a faster average reaction time than prior studies since it uses a novel technique to conduct the transactions.

[Updated][ Section 2: Literature Review, Paragraph 5, Page 3]

The searchable blockchain system can allow accurate search over encrypted distributed storage systems and protect privacy. Although it comes with limitations as only single-keyword searches are logically functional within the system and applying multi-keyword searches concerns confidentiality and efficiency of the database. S. Jiang et al.[18] suggested an advanced blockchain-based framework for information systems that can perform dynamic modifications and multi-keyword searches, ensuring preserved privacy. The bloom filter determines a low-frequency keyword from the database to operate search instructions providing reduced computational costs and search space. It also computes the operation in a single iteration, resulting in complete privacy protection. According to the outcomes, the multi-keyword search protocol exceeds the existing technique by 14.67 percent of time delay and 59.96 percent of financial expenses. Tuning the parameters of the filter system can bring in further improvements and provide a more robust system.

[Updated][ Section 2: Literature Review, Paragraph 6, Page 3]

Blockchain technology plays a vital role in securing the vast healthcare system. S. Jiang et al.[19] suggested a framework naming BlocHIE, a blockchain tool for sharing healthcare data. They analyze the sharing structure and store the data into two distinct blockchains EMR-Chain for electronic medical records and PHD-Chain for personal healthcare data. It abolishes the issues of the traditional storing system by employing on-chain verification, ensuring immutability and anonymity. Moreover, they apply FAIR-FIRST and TP&FAIR, two fairness-based packing techniques that increase system capacity and user fairness to ensure privacy-preserving.

[Updated][ References, Page 22]

  1. S. Jiang, J. Cao, H. Wu, and Y. Yang, “Fairness-based packing of industrial IoT data in permissioned blockchains,” IEEE Trans. Industr. Inform , vol. 17, no. 11, pp. 7639–7649, 2021.
  2. S. Jiang et al., “Privacy-preserving and efficient multi-keyword search over encrypted data on blockchain,” IEEE International Conference on Blockchain (Blockchain), 2019.
  3. S. Jiang, J. Cao, H. Wu, Y. Yang, M. Ma, and J. He, “BlocHIE: A BLOCkchain-Based Platform for Healthcare Information Exchange,” IEEE International Conference on Smart Computing (SMARTCOMP), pp. 49–56, 2018.

Reviewer 2, Concern # 5: Some terms should be defined on first use to broaden the scope of readers that could read this article in this journal. For example, Knapsack.

Author response: We would like to thank the honorable reviewer for the astute comments and suggestions. The missing insights that help readers broaden their scopes are now appropriately explained on first use. In the mentioned case, Knapsack 0/1 is briefly explained while stating its use in our experiment.

Author action: Modifications made to the updated manuscript such that readers can easily get a grasp of the mentioned terms.

[Updated][ Section 4.2.1.: Producer input selection , Paragraph 1, Page 10]

The system uses 0/1 Knapsack to determine the best possible value for electricity produced from an array of producers in the market. Knapsack derives a case in which the aim is to maximize the value in a knapsack while staying within the weight limitation. As per the functionality of 0/1 Knapsack, it considers either an entire item or rejects it altogether, given a collection of objects with certain weights and corresponding weights values. The algorithm takes the electricity produced per unit from the produced as weight and its corresponding wholesale price as the value of electricity. Moreover, the highest demand from customers is taken as the maximum weight capacity of the algorithm. Therefore, the maximum customer demand W is the knapsack capacity.

Reviewer 2, Concern # 6: I suggest a large amount of English language/grammar review, especially targeted at reducing the length of sentences to enhance clarity. For example, “To solve the dynamic pricing of a hierarchical energy market, this paper would like to propose a DR algorithm that copes up with the dynamic pricing and also helps to reduce the service provider’s (SP) and customer’s (CU) costs” (page 2). This sentence could be rewritten as “To help with dynamic pricing in a hierarchical energy market, this paper proposes a DR algorithm that keeps up with the dynamic pricing while also reducing the service provider’s (SP) and customer’s (CU) costs.”

Author response: We would like to express our deepest gratitude to the honorable reviewer for the valuable suggestion. While proofreading, the needed grammatical changes are made as well as proper sentence structures are used to ensure there is adequate clarity in the topics. During such changes the length of the sentences are reduced and structured properly to improve readability.

Author action: We have updated Section 1: Introduction and fixed the language issues.

[Updated][Section 1: Introduction, Paragraph: 2, Page 2]

Demand response (DR) is an efficient method to use in smart grid system to reduce cost and improve grid efficiency. Price based DR refers to influencing the customers’ electricity usage with the variable electricity prices. To help with dynamic pricing in a hierarchical energy market, this paper proposes a DR algorithm that keeps up with the dynamic pricing while also reducing the service provider’s (SP) and

customer’s (CU) costs. With the help of Reinforcement Learning (RL) and Q-Learning we are predicting the price of electricity. The SPs get the flexibility to set the price dynamically with accordance to demand and level of dissatisfaction. Furthermore, blockchain integration to every customer’s profile will help to secure the decentralized transaction of electricity between SP and CU.

Reviewer 2, Concern # 7:  Further, “A smart contract is a bunch of rules and agreements approved by both parties.” Could be “a set of rules…”.

Author response: We would like to thank the honorable reviewer for the astute comments and suggestions. In the revised manuscript, we have used “a set of rules” instead of “a bunch of rules” in the mentioned sentence. Additionally, proper measures are taken to ensure the rectification of grammatical errors and improved readability. The updated manuscript contains English that is precise and easily readable.

Author action: Modifications made to the manuscript in this regard is as follows:

[Updated][Section 3.2:Smart Contract, Paragraph: 1, Page 5]

A smart contract is a set of rules and agreements approved by both parties. It offers authenticity to verify the effectiveness of any operation or action. After deploying the smart contract, some functions and events are performed to confirm the transaction. It can assist us in transferring money, transferring assets and shares, and other significant transactions in a transparent manner.

Reviewer 3 Report

The manuscript presents a blockchain-based smart grid system.
The topic is of current interest.
However, there are some questions as follows:
- In the majority of the manuscript, the term "blockchain" is used. However, in the title and abstract, "block-chain" is used. The authors are recommended to make the terms consistent.
- It's good to summarize the contributions in introduction. However, some of the descriptions are too wordy. Please try to make the main contributions concise and clear.
- Some important related works are missing, e.g., "Fairness-based packing of industrial IoT data in permissioned blockchains", "Blochie: a blockchain-based platform for healthcare information exchange" and "Privacy-preserving and efficient multi-keyword search over encrypted data on blockchain".
- Many figures are blurred. Can they be replaced by vector-based figures?
- Q-learning has been developed for decades. What are the new parts in the developed Q-learning algorithm?
- Ethereum employs Proof of Work as the consensus mechanism. Will it cost too much energy, especially considering the smart grid application scenario?

Round 2

Reviewer 1 Report

The manuscript improved a lot, it can be published in this form.

Reviewer 3 Report

All my comments are well addressed.